# Predicting Progression of Autosomal Dominant Polycystic Kidney Disease by Changes in the Telomeric Epigenome

**DOI:** 10.3390/cells11203300

**Published:** 2022-10-20

**Authors:** Ismail Kocyigit, Serpil Taheri, Cihan Uysal, Mehmet Memis, Salih Guntug Ozayturk, Gokmen Zararsiz, Minoo Rassoulzadegan

**Affiliations:** 1Department of Nephrology, Medical School, Erciyes University, 38280 Kayseri, Turkey; 2Betul Ziya Eren Genome and Stem Cell Center, Erciyes University, 38280 Kayseri, Turkey; 3Department of Medical Biology, Medical School, Erciyes University, 38280 Kayseri, Turkey; 4Department of Internal Medicine, Medical School, Erciyes University, 38280 Kayseri, Turkey; 5Department of Biostatistics, Medical School, Erciyes University, 38280 Kayseri, Turkey; 6INSERM-CNRS, Université de Nice, 06107 Nice, France

**Keywords:** polycystic kidney disease, genome integrity, epigenetics, lncRNA, telomere, *TERRAs*, R-loop

## Abstract

Autosomal dominant polycystic kidney disease (ADPKD) is the most common inherited cause of chronic kidney disease with *Polycystin* (*PKD*) 1 and 2 gene mutation. However, the intra-familial variability in symptoms further suggests a non-Mendelian contribution to the disease. Our goal was to find a marker to track the epigenetic changes common to rapidly progressing forms of the disease. The risk of ADPKD increases with age, and aging shortens the telomere length (TL). Telomeres are a nucleoprotein structure composed mainly of three complexes, shelterin, CST and *RNA-containing telomere repeat*
*(TERRA)*, which protects the ends of chromosomes from degradation and fusion, and plays a role in maintaining cellular stability and in the repair of telomeric damage. *TERRAs* are transcribed from telomeric regions and a part of them is engaged in a DNA/RNA hybrid (R-loop) at each chromosome end. We tracked TL and *TERRA* levels in blood samples of 78 patients and 20 healthy control. Our study demonstrates that TL was shortened and *TERRA* expression levels in the DNA-attached fraction increased in autosomal dominant polycystic kidney patients with mutations in *PKD1* and *PKD2* compared to the control group. Moreover, it was observed that the expression of *TERRA* engaged in the R-loop was higher and the length of telomeres shorter in patients with ADPKD who showed rapid disease progression. Intrafamilial variation in TL and *TERRA* levels with the same mutation would indicate reliable epigenetic potential biomarkers in disease monitoring.

## 1. Introduction

Autosomal dominant polycystic kidney disease (ADPKD) is the most common inherited cause of end-stage renal disease. Approximately 85% of patients have mutations in the *Polycystin 1 (PKD1)* gene and 15% in the *Polycystin 2 (PKD2)* gene [1]. The role of the *PKD1* and *PKD2* gene mutations in the pathogenesis of ADPKD is known. The clinical hallmark of ADPKD is bilateral renal enlargement, due to numerous cysts. The cysts are lined with hyperproliferative and hypersecretory epithelial cells, which exhibit impaired cellular metabolism [2]. Many signaling pathways, such as cAMP, mTOR and cMyc, are known to play a role in ADPKD [3]. However, there are differences in the clinical progression of family members with the same genetic mutation [4]. These differences further suggest genetic and epigenetic changes associated with the progression of renal cysts [5,6]. Alterations in DNA methylation and histone modifications at certain loci and cell cycle dysregulation have also been reported in the pathogenesis of ADPKD [3]. Recent studies have also suggested that increased inflammation and oxidative stress during ADPKD are correlated with hypertension and cardiovascular damage [6,7].

Telomere attrition under stress, such as chronic inflammation and oxidative damage, is linked to the guanine-rich sequence in the telomere hexameric tandem repeat DNA sequences (TTAGGG) at the ends of the chromosomes. Telomeres are replicated during the cell cycle, and the last part of the lagging strand is filled in by telomerase (a reverse transcriptase). Additionally, the limited activity of the enzyme telomerase in most somatic cells causes telomere shortening, which ultimately leads to replicative senescence (cell cycle arrest), a well-known physiological process during aging [8,9]. The age-dependent development of ADPKD motivated this study to clarify whether markers of genomic stability, such as telomere length and its transcribed regions related to genetics and /or epigenetics, are simultaneously altered in patients with a high risk of ADPKD development.

In addition, telomeres are protected by the structure of nucleoproteins: shelterin (TRF1, TRF2, TIN2, RAP1, POT1 and TPP1), CST (CDC13/CTC1, STN1 and TEN1) and RNA-containing telomeric repeat. Shelterin protects telomeres from the DNA damage response (DDR), while CST and telomeric RNA regulate telomere extension by telomerase [9]. 

Telomeric RNAs (UUAGGG repeats) associated with telomeric repeats (*TERRAs*) are long non-coding transcripts [9,10]. *TERRAs* are transcribed by RNA polymerase II [11,12] from the telomere of each chromosome, and part of the nascent transcripts remains associated with telomeres in the form of hybrid DNA/RNA structures (R-loop) in normal cells [10,11]. *TERRAs* ensure genome stability by regulating telomerase activity, telomere length and heterochromatin maintenance. Variations in the *TERRA* transcripts are associated with variable telomere length [9], uncontrolled cell division, genome instability and cellular senescence. *TERRAs* bind to Telomerase Reverse Transcriptase (TERT), thereby inhibiting telomerase activity [13]. Another task of *TERRAs* is to regulate the transcription of genes located near their binding sites [14]. In conclusion, the dynamic balance of *TERRA* appears to have a critical role in controlling the telomere length at all stages of the cell cycle [12].

Telomere length (TL) is tightly protected and any damage leads to uncontrolled cell division or premature aging [12]. Moreover, TL is shortened with each round of DNA replication during somatic cell division and could also be altered by environmental factors such as increased oxidative stress. The shortening of telomere length with age or during various pathologies is associated with the worsening of the disease [7,8].

We focused our study on TL and *TERRA* levels from 78 blood samples from patients with mutations in the *PKD1* and *PKD2* genes or from controls. In fact, our results evidence the relationship between ADPKD (age-dependent risk) with telomere shortening and higher levels of *TERRA* retained on chromosomes. Among the 78 patients, sixteen individuals from distinct families with ADPKD rapid development profiles showed higher levels of *TERRA* as a DNA/RNA hybrid at the ends of chromosomes with shorter telomeres. Another twenty-four members of eleven slowly evolving families with *PKD1* or *2* mutations, but with differences in the variation of TL and *TERRA* levels within the same family, suggest epigenetic changes in addition to the same mutation.

## 2. Materials and Methods

Seventy-nine patients diagnosed with ADPKD and 20 healthy individuals for the control group were enrolled in this study at the Erciyes University School of Medicine, Department of Nephrology. Permission for this study was obtained from the Erciyes University Ethics Committee, with decision number 11.03.2020 and 2020/176. Telomere length and hybrid *TERRA* levels in blood samples taken from patients and controls were determined by the Erciyes University Betul-Ziya Eren Genome and Stem Cell Center (GenKok) Genome Unit.

### 2.1. Materials

PureZOL™ RNA isolation solution (Bio-Rad, 7326890, Hercules, CA, USA), EvoScript Universal cDNA Master (Roche, 7912374001, Mannheim, Germany), Light Cycler 480 SYBR Green Master Mix (Roche, 4707494001, Mannheim, Germany), Plasmid DNA: pGL3 Luciferase Reporter Vectors (Promega-E175A, Madison, WI, USA), Integrated DNA Technologies (IDT, Newark, NJ, USA) brand primers, Nuclease-Free Water (Qiagen, 129115, Hilden, Germany), 1.5 mL and 2 mL Eppendorf tubes and Roche Light Cycler^®^ 480 II Plate (Roche, 04729692001, Mannheim, Germany) were used as materials.

### 2.2. Study Population

The registration method for the patients included in the study was prospective, and 78 patients aged between 18 and 75 applied to the Nephrology Department of the Faculty of Medicine of the Erciyes University. Patients diagnosed with ADPKD and having no other chronic disease besides ADPKD were included. The patients included in the study had mutations in the *PKD1* or *PKD2* genes.

The definition of rapid progression may vary by country or region. However, a general definition of progression can be used, which includes a change in eGFR or albuminuria category, or both, as well as numerical changes over an established period of time. Rapid progression was defined as a sustained decrease in eGFR of more than 5 mL/min per 1.73 m^2^ per year or 2.5 mL/min/1.73 m^2^ per year over 5 years, according to KDIGO recommendations [15].

The study included 20 healthy individuals selected from volunteers who applied to Erciyes University School of Medicine for joint control and were matched with the patient group used in terms of age and of gender.

### 2.3. DNA Isolation from Blood Samples

Venous blood was collected from patients and controls in sterile EDTA in 4-mL tubes. Genomic DNA was isolated from whole blood using High Pure PCR Template Preparation Kits (Roche Diagnostics, GmbH, Mannheim, Germany). Half of the isolated DNA sample was used to determine the telomere length, and the remaining part was used for DNA:RNA hybrid isolation to determine *TERRA* levels.

### 2.4. Determination of Telomere Length

Telomere lengths were determined by PCR in a Light Cycler 480 II (Roche, Mannheim, Germany) device using IDT (Integrated DNA Technologies, Coralville, IA, USA). Primers and their sequences are shown in Table 1.

Following PCR, using the Ct values obtained for the telomere standard and 36B4 standard, a standard curve graph was drawn, and telomere lengths were calculated using the method of Callaghan et al. [16].

### 2.5. DNA: RNA Hybrid Isolation from DNA Samples

Two-milliliter peripheral venous blood samples were collected from the participants in a tube containing EDTA; after isolation of DNA from blood, isolation of RNA from DNA:RNA hybrid was achieved from the DNA samples by extensive *DNase* treatment. The RNAs released from the DNA were then purified by standard centrifugation and washing steps; see below.

DNA dissolved in Buffer AE was centrifuged at 12,000× *g* for 10 min by adding the same volume of isopropanol as the sample volume. After centrifugation, the aqueous phase was discarded, and 1 mL of a 70% EtOH solution was added to the pellet and centrifuged at 7500× *g* for 5 min. After centrifugation, the aqueous phase was discarded, and the alcohol in the tube was removed. Then, 10 µL of DNase + 90 µL of *DNase* incubation buffer was added to the pellet and kept at room temperature for 15 min. Then, 300 µL of Trizol (Qiagen, 79306, Hilden, Germany) and 180 µL of chloroform were added, vortexed and centrifuged at 12,000× *g* for 20 min. After centrifugation, the aqueous phase was taken up in a new tube; an equivalent volume of isopropanol was added to the aqueous phase and centrifuged at 12,000× *g* for 10 min. After centrifugation, the aqueous phase was discarded, and 1 mL of a 70% EtOH solution was added to the pellet and centrifuged at 7500× *g* for 5 min. After centrifugation, the aqueous phase was discarded, and 1 mL of a 70% EtOH solution was added to the pellet and centrifuged at 7500× *g* for 5 min. Then, 30 µL of nuclease-free water was added to the pellet. The obtained DNA and RNA from hybrid structure samples were stored at −80 °C until the start of the experiments.

### 2.6. cDNA Synthesis

To determine *TERRA* levels from DNA:RNA hybrid samples obtained from participants’ DNA samples, an EvoScript Reverse Transcriptase cDNA synthesis (07912374001, Roche, Mannheim, Germany) kit was used according to the manufacturer’s protocol.

The reaction mixture was prepared in the amounts according to the kit protocol for cDNA isolation. Then, 5 µL of RNA was added to each tube, and 2 µL of reverse transcriptase enzyme was added to each sample. Next, the desired product was placed in the Thermal Cycler device under appropriate reaction conditions for PCR. At the end of PCR, 20µL of cDNA product was formed and cDNA samples were diluted 1:5 with nuclease-free water.

### 2.7. Determination of DNA:RNA Hybrid TERRA Levels in Real-Time PCR

*TERRA* transcript levels were determined from DNA:RNA hybrid RNA samples in the Roche Light Cycler LC480 II (Roche, Mannheim, Germany) using the LightCycler^®^ 480 SYBR Green I Master Kit (Roche, Cat. No. 04707516001, Mannheim, Germany).

Real-time PCR conditions consisted of an initial denaturation step at 95 °C for 5 min, followed by 40 cycles of 94 °C for 20s, 60 °C for 20s, 72 °C for 45 s and 95 °C for 15 s, and, finally, a melting curve was performed at 67 °C for 1s (melting curve) and cooled at 40 °C for 30 s.

The Ct values of the samples were determined in a Roche Light Cycler LC 480 II Real-Time PCR setup. All samples were run in duplicate to rule out manipulation errors, and Ct values were averaged. β-Actin was used as the house-keeping gene. Ct values obtained at the end of PCR were normalized using the 2^−ΔΔCt^ method [17,18].

### 2.8. Statistical Analyses

Histograms, q–q plots and Shapiro–Wilk’s test were applied to assess the data normality. The Levene test was used to test variance homogeneity. To compare the clinical parameters between normal and rapid progressor disease groups, a two-sided independent-samples *t*-test or Mann–Whitney U test was used for continuous variables, while the Fisher–Freeman–Halton test was used for categorical variables. Univariate and multiple binary logistic regression analyses were applied to identify the risk factors of rapid progression in ADPKD patients. Significant variables at the *p* < 0.25 contingency level were included in multiple models, and forward elimination was performed using Wald statistics to identify the independent risk factors of rapid progression. The model’s goodness-of-fit was assessed using the Hosmer–Lemeshow test and Nagelkerke R^2^ statistic. Moreover, a receiver-operating characteristic curve (ROC) analysis was used to assess the diagnostic effect of *TERRA* expression on rapid progression. Furthermore, the area under the ROC curve was calculated with 95% confidence intervals. The Youden index was used to identify the optimal cut-off value. Sensitivity, specificity, positive and negative predictive values and positive and negative likelihood ratio statistics were calculated with 95% confidence intervals based on the identified cut-off values. Analyses were conducted using TURCOSA (Turcosa Analytics Ltd. Co, Kayseri, Turkey, www.turcosa.com.tr) statistical software.

## 3. Results

Our study included 78 patients diagnosed with ADPKD with *PKD1* or *PKD2* gene mutations and 20 healthy individuals. In this group of patients, 62 of the patients were classified as normal regressors, while 16 of the patients from distinct families were classified as rapid progressors. The participants were 56% men, with a mean age of 48 ± 13 years, and 44% women, with a mean age of 46 ± 11 years. The mean age of the 20 control individuals was 47 ± 10 years (55%) for the men and 47 ± 10 years (45%) for the women. There was no difference between patients and control groups regarding age and sex. Based on mutations and clinical parameters, the relationship between telomere length and levels of *TERRA* as DNA:RNA hybrid was determined by qRT-PCR.

Comparing the telomere lengths of ADPKD patients and the control group, a significant shortening of telomere length was found in the ADPKD patient group compared to the control group (*p* < 0.0001) (Figure 1A and Table 2).

*TERRA* transcripts are present in cells in the form of free RNA and in minute quantities in the form of hybrid DNA:RNA (R-loop). The amount of *TERRA* in the R-loop varies with telomere size. We hypothesized that the assessment of the amount of *TERRA* retained on genomic DNA should be more informative in response to pathological conditions. To extract *TERRA* retained on the genome—see Methods section—briefly, blood cells are lysed with Trizol–chloroform buffer, and then the aqueous-phase RNA and genomic DNA at the interface of phenol/chloroform are purified. From the DNA fraction, the RNAs retained in the form of a hybrid DNA:RNA structure (R-loop) are extracted by extensive treatment with *DNase*. The lengths of telomeres and the amounts of *TERRA* were evaluated by qRT-PCR in the DNA:RNA fraction.

On the comparison of the levels of *TERRAs* retained on the genomes (DNA:RNA hybrid) of the ADPKD patients (see Methods) and the control group, the levels of *TERRA* proved to be significantly higher in the ADPKD patients than in the control group (*p* = 0.0009) (Figure 1B).

Additionally, ADPKD patients and control groups were separated by gender, and the telomere length and *TERRA* levels were compared with each other, but no significant differences were found. The results are shown in Figure 1C,D (lower panels).

When telomere lengths and *TERRA* levels were compared in ADPKD patients with *PKD1* or *PKD2* gene mutations, there was no difference in telomere length between mutation types (Figure 2A,B and Table 2). However, hybrid *TERRA* (DNA:RNA) levels were detected at higher levels in patients with a *PKD1* gene mutation; the results are summarized in Table 3 and Figure 2C,D (lower panels).

Variables of CRP, proteinuria and hybrid *TERRA* levels in the multiple models predicted 74.5% of the variability in rapid progression; the results of logistic regression and descriptive analysis are summarized in Table 4. In multiple binary logistic regression analysis, the ORs (95%CI) of CRP, proteinuria and *TERRA* levels were 1.21(1.05–1.38), 16.64(1.13–245.47) and 5.60(1.46–21.51), respectively (Table 4). The Hosmer–Lemeshow test resulted in *X*^2^ = 6.247, *p* = 0.620. These results demonstrated the relevance of the multiple binary logistic regression model to predict the rapid progression of patients with ADPKD.

In agreement with the results that we present, the higher proportions of *TERRA* retained in hybrid with DNA (genome of ADPKD patients) are a predictive factor of rapid progression; the results are presented in Figure 3 and Table 5. The Nagelkerke R2 statistic was calculated as 0.745 (Equation (1)).
Probability of rapid progression = 1+ e^(−7.601+0.187Crp+2.812Proteinuria+1.723TerraExpression)^(1)

Equation (1) Nagelkerke R2 statistic

All ADPKD patients with rapid progression stand out from the rest due to the shorter telomere length (*p* = 0.041) and higher levels of *TERRA* in hybrid structures with DNA (*p* < 0.001) compared to those with slow progression. The comparative analysis and results are shown in Figure 4 A,B. ADPKD patients with *PKD1* mutation and rapid progression stand out from the rest due to the shorter telomere length (*p* = 0.018) and higher levels of *TERRA* in hybrid structures with DNA (*p* < 0.001) compared to those with slow progression. The comparative analysis and results are shown in Figure 4C,D. Since the number of rapidly progressing patients with *PKD2* mutation was only one, a statistical comparison could not be made. However, the ADPKD patient with the *PKD2* mutation and rapid progression had a shorter telomere length and higher levels of *TERRA* in hybrid form with DNA compared to patients with slow progression (Figure 4D,E).

Finally, we compared members of the same family in the cohort. Twenty-four patients came from eleven distinct families (Figure 5A). Members of the same family carrying the same *PKD 1* or *2* mutation showed variation in telomere length and *TERRA* levels, independently of gender and age (Figure 5B).

## 4. Discussion

Patients with ADPKD are classified according to the rate of slow or rapid disease progression. The rapid progression of ADPKD leads to renal failure around the age of 50, with an as-yet-undetermined threshold. Anticipating rapid progress is essential to assessing the benefit/risk ratio of any intervention and adopting long-term renal protection approaches in early ADPKD [19]. Therefore, we propose reliable potential biomarkers to predict kidney function decline.

Here, we show that patients with *PKD1* and/or *PKD2* gene mutations also carry functional alterations in *TERRA* lncRNA level profiles in the DNA:RNA hybrid fraction. In fact, we found that in the blood cells of patients with ADPKD, lncRNAs such as *TERRAs* are more retained on the DNA and the telomere length (TL) is shorter. Indeed, at each cell cycle before DNA synthesis, the R-loops are resolved by RNase H [11] to allow the replication fork to move forward to copy a new strand. Here, we reveal particular changes in functional structure as DNA:RNA hybrids are known to be under the precise control of genome surveillance. Our studies show that blood cells from ADPKD patients shorten the telomere and accumulate *TERRA* in the DNA:RNA hybrid structure.

Renal failure due to the development of parenchyma cysts in the kidneys, as well as the onset of hypertension at an early age, is a prominent feature of ADPKD. Although essential hypertension occurs in the fifth decade in the normal population, hypertension develops in the third decade in patients with ADPKD. The average age at the beginning of dialysis is 50 years old in ESRD with *PKD1* mutation, whereas the average age is 70 years with *PKD2* mutation. Therefore, the existence of additional factors (non-Mendelian) has been considered to contribute to the formation of hypertension and the progression to ESRD [20]. In recent studies, ADPKD was characterized by early inflammation, oxidative stress, hypertension and cardiovascular involvement [21]. Moreover, differences in the clinical progression of family members with the same genetic mutation suggest underlying mechanisms in addition to genetic alterations [4].

Kidney aging is a complex, multifactorial process characterized by many anatomical and functional changes, and various factors play a role in kidney aging, such as loss of telomeres, cell cycle arrest, chronic inflammation, activation of the renin–angiotensin system, decreased expression and development of klotho and tertiary lymphoid tissues. These changes could also be seen in many other types of kidney disease. Moreover, many aging-related comorbidities have been hypothesized to trigger the acceleration of renal aging [22]. To develop new therapeutic strategies for premature renal aging, it is important to elucidate the mechanisms underlying renal aging and to identify molecular alterations.

A recent Mendelian randomization study supports the causal link between decreased telomere length and impaired kidney function [23,24]. Telomere length is known to be inversely proportional to aging, which has been proposed as a marker for age-related diseases. Telomere shortening can be accelerated by oxidative stress and inflammation, which commonly occur in CKD and ADPKD. In the leukocytes of patients with chronic kidney disease (CKD), telomeres are shorter [23,24,25]. Here, we report shorter telomeres in the whole blood cells of ADPKD patients with mutations in the *PKD1* or *PKD2* gene. More importantly, engaged *TERRA* levels in DNA:RNA hybrids increased with telomere shortening in patients with ADPKD. The higher risk in men due to the difference in cellular metabolism has been put forward as a reason for the accelerated progression of kidney disease [2,4]. However, no gender differences were observed in the expression levels of *TERRA* and TL in the ongoing study group.

### Shorter Telomere

The relationship between telomere length and mortality was studied in a large cohort of patients with CKD and high proteinuria, which included 4955 patients with stage G3 and A1-3 or G1-2 and A3. A follow-up period was carried out for four years and 354 deaths were detected. The study model was adjusted for age, sex, baseline eGFR, urine albumin to creatinine ratio, diabetes mellitus, prevalent cardiovascular disease, LDL-cholesterol, HDL-cholesterol, smoking, body mass index, systolic and diastolic blood pressure, C-reactive protein and serum albumin. Physiological and psychological stressors also lead to faster telomere shortening [23]. The additional effects of reactive oxygen species (ROS) and cortisol, which occur with stress in the aging process, and telomeres, have been previously reported [22]. Therefore, telomere length was determined to be a strong and independent predictor of all-cause mortality. Accordingly, a 0.1 unit decrease in relative telomere length was associated with a 14% increased risk of death [25,26].

In studies conducted after the discovery of *TERRA*, it was determined that *TERRA* participates in the protection of the terminal regions of the chromosomes and the stabilization of the genome. After having been transcribed from the sub-telomeric regions, parts of the *TERRA* molecules are found in hybrid DNA:RNA at the telomeres [27]. However, it has not been fully elucidated how telomeres and *TERRA* play a role in the aging process and disease pathogenesis. There are very few studies in the literature on how *TERRA* works. In normal cells, part of the nascent *TERRA* transcript remains at each telomere as a DNA:RNA hybrid [11,28]. In certain pathological conditions, *TERRA* molecules accumulate on the short telomere [10,29].

DNA:RNA hybrids are three-stranded nucleic acid structures. It has been shown that the accumulation of DNA:RNA hybrids can generate homologous recombination [13]. In the presence of telomerase and homologous chromosomes, increased *TERRA* accumulation in short telomeres promotes telomere elongation through homologous recombination, thereby maintaining telomere length [29]. *TERRA* has also been suggested to prevent cellular senescence by promoting homologous recombination at native levels of extremely short telomeres in DNA hybrids [13]. It is proposed that the high level of *TERRA* in hybrids on telomeres is a response to stress [30]. It thus becomes essential to determine the functional mechanisms of telomeres and *TERRA* in many diseases under different stress conditions [13,28].

In the analysis of subgroups, telomere length shortening and higher hybrid *TERRA* levels were detected in the rapid progression group than in the normal progression group. Additionally, when the telomere lengths of patients with subsets of *PKD1* and *PKD2* gene mutations were compared, there was no difference in telomere length between patients. In contrast, higher levels of *TERRA* as a DNA:RNA hybrid were detected on the genomes of patients carrying *PKD1* gene mutations.

The *PKD1* gene encodes an integral membrane protein that functions as a regulator of calcium-permeable cation channels and intracellular calcium homeostasis [31,32]. It is also involved in cell–cell/matrix interactions and may modulate G-protein-coupled signal transduction pathways and plays an important role in renal tubular development. The *PKD1* gene is located near the 16p telomere region. The terminal region of human chromosome 16p can be subdivided into two GC-rich, Alu-rich domains and one GC-rich, Alu-poor domain. This region is also rich in genes that play important roles in maintaining genome integrity, essential in cellular processes such as DNA recombination, repair, transcription, RNA processing, signal transduction, intracellular signaling and mRNA translation [33]; see listing in Appendix A. The increased level of *TERRA* transcripts in patients with *PKD1* gene mutations may be due to the mutational position effects.

The *PKD2* gene is located in the 4q22.1 region and codes for a member of the polycystin protein family. The *PKD2* protein is a multi-pass membrane protein that functions as a calcium-permeable cation channel and is involved in calcium transport and calcium signaling in renal epithelial cells. This protein interacts with polycystin 1 and is involved in a common signaling cascade in tubular morphogenesis [34]. The implications of *PKD2* gene products in maintaining telomere and genome integrity is not known, but our results indicated a role in patients with *PKD2* mutations.

Do *PKD1* and *PKD2* mutations cause conditions that lead to altered telomere stability? It is not yet clear whether *PKD1* and *PKD2* mutations induce a variation in the expression of global non-coding transcripts such as *TERRAs*, which leads to shorter telomeres or to *TERRA* levels that are increased due to telomere damage. Transcriptomic analysis of the mouse model of ADPKD with *PKD1* mutations of the rapid cyst-forming model, and human ADPKD, revealed a number of altered transcripts in cyst formation and suggests that metabolic changes may play a role in ADPKD and may accelerate disease progression [35,36]. Unfortunately, there is no transcriptomic analysis of blood cells from patients with *PKD1* or *PKD2* mutations yet. Blood cells can be expected to represent a general profile of cellular transcripts in patients with ADPKD. Altered metabolic changes in the cells of ADPKD patients can lead to altered transcripts in the blood and thus general instability, which requires further investigation with functional telomere damage as a side effect. It is plausible that alterations in transcription may be the initial signals and that the increased retention of *TERRA* on DNA will lead to telomere shortening.

A limitation of this study is that the activity of *TERRA* was only tested in our current population of 78 patients and there were no additional test data. Larger studies are needed to determine thresholds for *TERRA* levels as a potential biomarker. In contrast, intrafamilial (eleven distinct families) variations in TL and *TERRA* levels with the same mutation, as revealed here, suggest that epigenetic changes contribute to the progression of ADPKD.

## 5. Conclusions

Therefore, in this study, telomere length and *TERRA* levels in ADPKD as DNA:RNA hybrids reveal clues in detecting disease progression. We believe that telomere length and, in particular, the use of *TERRA* levels as a potential biomarker may be reasonable in predicting the progression of ADPKD.

## Figures and Tables

**Figure 1 cells-11-03300-f001:**
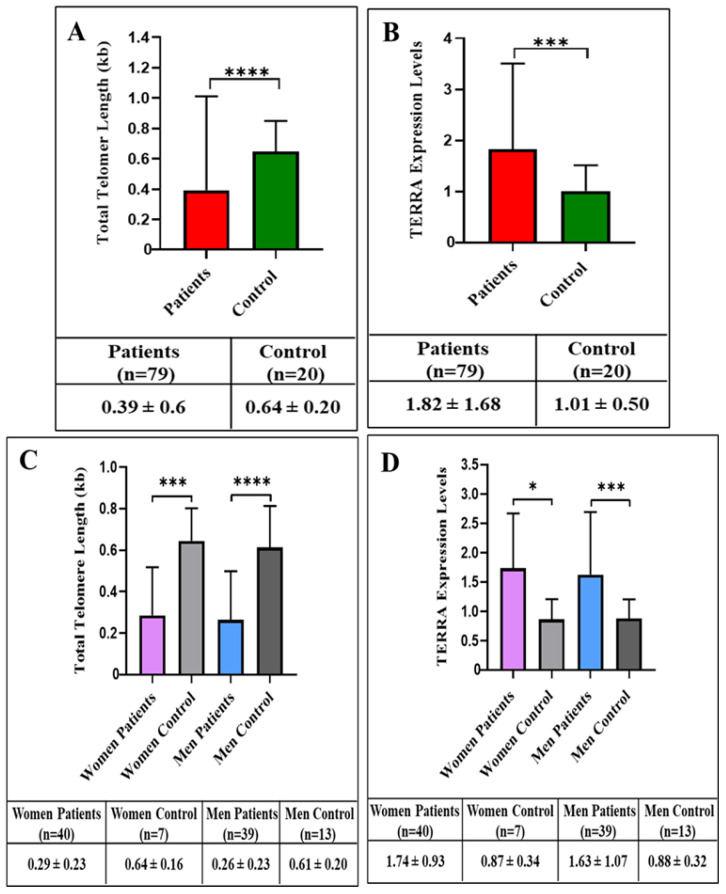
Determination of telomere length (PCR; see Methods) from blood samples from the group of patients diagnosed with ADPKD and from the control group. Telomere length is significantly shorter in patients than in controls (**** *p* < 0.0001) (**A**). *TERRA* expression levels in patients and control groups. *TERRA* expression levels were determined in the DNA:RNA hybrid fraction (see Methods) extracted from blood cells. Significant differences in *TERRA* level were found in patients and controls (*** *p* = 0.0009) (**B**). Telomere lengths were compared between patient and control groups by gender. Telomere length of female (*** *p* = 0.0004) and male (**** *p* < 0.0001) patients was found to be shorter than that of controls. (**C**) *TERRA* expression levels were compared between patient and control groups by gender. *TERRA* expression levels in female (* *p* = 0.0102) and male (*** *p* = 0.0004) patients were found to be increased compared to controls. All data are plotted as mean ± standard deviation (**D**).

**Figure 2 cells-11-03300-f002:**
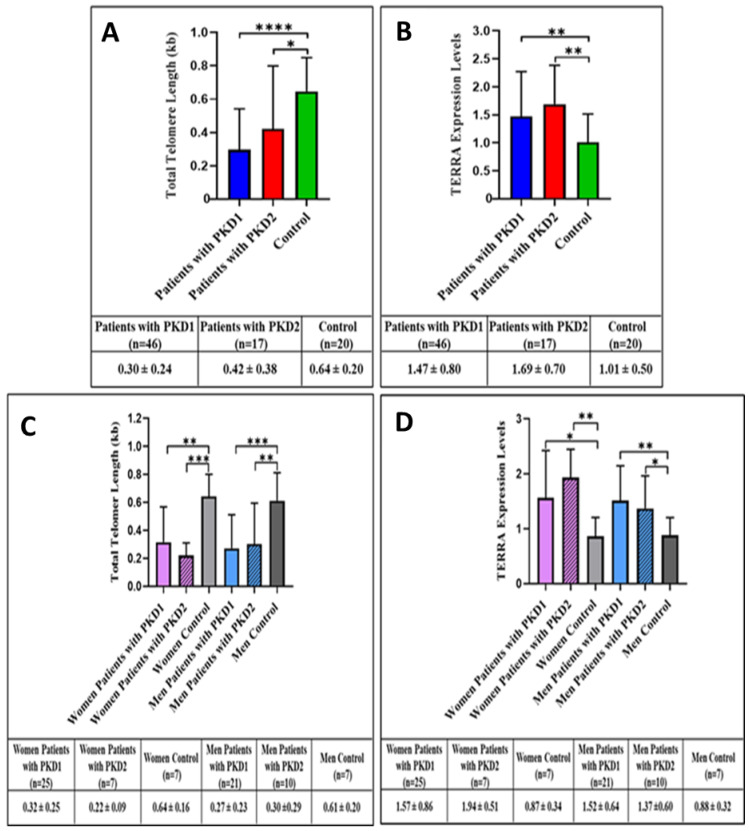
Telomere lengths were compared between patients carrying the *PKD1* and *PKD2* mutations, and the control group. The telomere length of *PKD1* (**** *p* < 0.0001) and *PKD2* patients (* *p* = 0.035) was found to be shorter than that of the control group. (**A**) *TERRA* expression levels were compared between patients carrying the *PKD1* and *PKD2* mutations, and the control group. Expression levels of *TERRA* of *PKD1* (** *p* = 0.0066) and *PKD2* patients (** *p* < 0.0015) were found to be increased compared to controls. (**B**) Telomere lengths were compared between women and men with the *PKD1* or *PKD2* mutation, and control groups by gender. The telomere length of women with *PKD1* (** *p* = 0.0022) and with *PKD2* (*** *p* = 0.0003) was found to be shorter than that of the control group. The telomere length of men with *PKD1* (*** *p* = 0.0003) and men with *PKD2* (** *p* = 0.0029) with was found to be shorter than that of the control group. (**C**) The expression levels of *TERRA* were compared between women and men carrying the *PKD1* and *PKD2* mutations, and the control groups according to gender. *TERRA* expression levels of female patients with *PKD1* (* *p* = 0.0324) and *PKD2* (** *p* = 0.0043) were found to be increased compared to controls. *TERRA* expression levels of male patients with *PKD1* (** *p* = 0.0044) and male patients with *PKD2* (** *p* = 0.0305) were found to be increased compared to controls. All data were plotted as mean ± standard deviation (**D**).

**Figure 3 cells-11-03300-f003:**
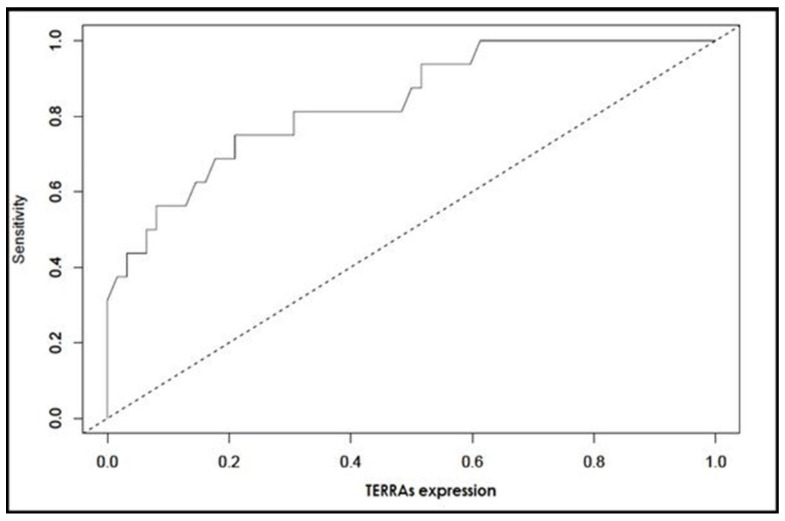
ROC analysis showing the diagnostic impact of *TERRA* levels in DNA:RNA fraction on rapid progression.

**Figure 4 cells-11-03300-f004:**
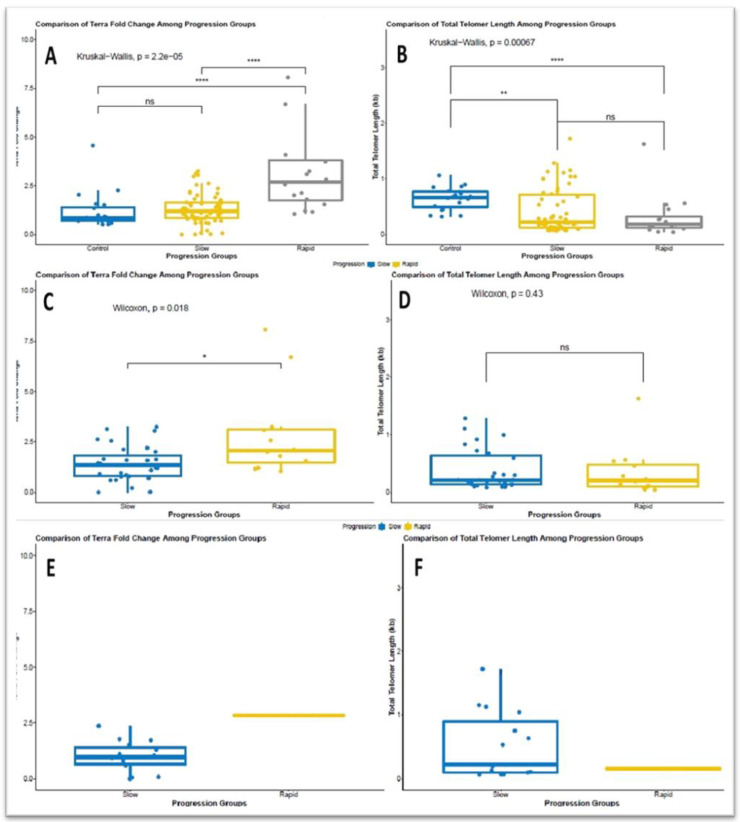
Assessment of normal and rapid progression in terms of *TERRA* levels extracted from the DNA:RNA hybrid fraction in all ADPKD patients compared to controls (**** *p* = 2.2^−5^) (**A**). Assessment of normal and rapid progression in terms of telomere length in all ADPKD patients compared to controls (** *p* = 0.000067) (**B**). Assessment of normal and rapid progression in terms of *TERRA* levels extracted from the DNA:RNA hybrid fraction in ADPKD patients with *PKD1* mutation(* *p* = 0.018) (**C**). Assessment of normal and rapid progression in terms of telomere length in ADPKD patients with *PKD1* mutation(ns, *p =* 0.43) (**D**). Assessment of normal and rapid progression in terms of *TERRA* levels extracted from the DNA:RNA hybrid fraction in ADPKD patients with *PKD2* mutation (**E**). Assessment of normal and rapid progression in terms of telomere length in ADPKD patients with *PKD2* mutation (**F**).

**Figure 5 cells-11-03300-f005:**
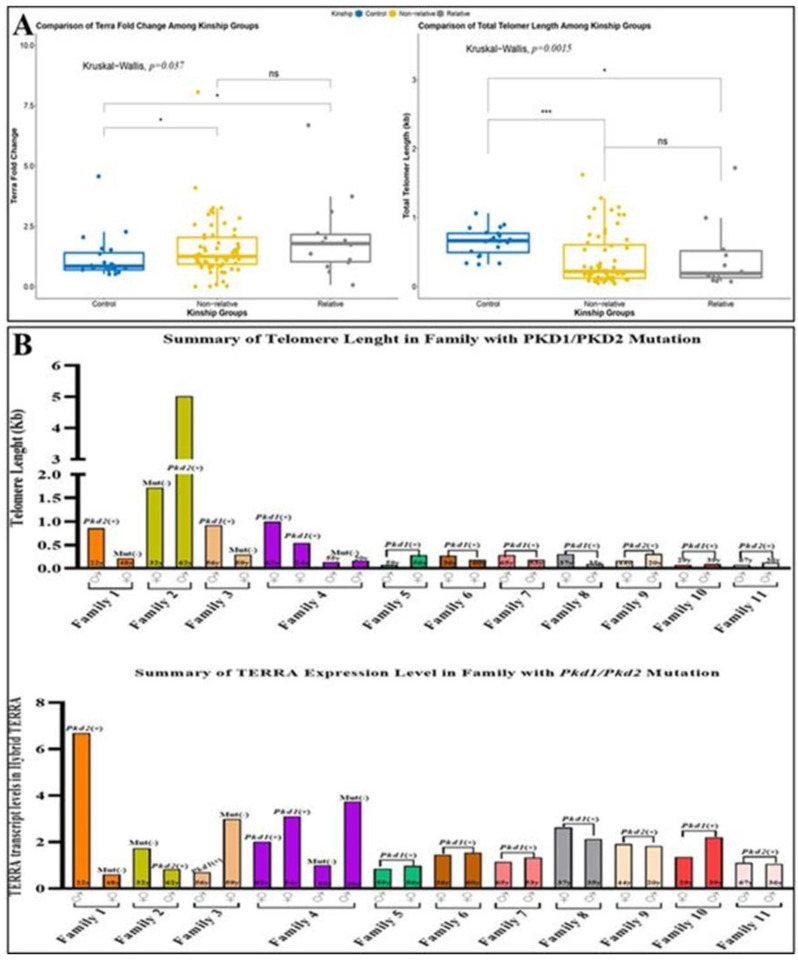
Assessment of relative and nonrelative ADPKD patients in terms of *TERRA* levels and telomere lengths *(* p =* 0.037, **** p =* 0.0015) (**A**). Comparison of telomere lengths and hybrid *TERRA* levels of individuals among ADPKD families according to *PKD1/2* mutations (**B**).

**Table 1 cells-11-03300-t001:** Primers and their sequences used to determine telomere length [16].

Oligomers		Species	Oligomer Sequences (5′→3′)	Amplicon Size
**Standards**	Telomer standard	human	(TTAGGG)_14_	84 bp
	36B4 standard	Human	CAGCAAGTGGGAAGGTGTAATCCGTC TCCACAGACAAGGCCAGGACTCGTTTGTACCCGTTGATGATAGAATGGG	>75 bp
**PCR Primers**	Telo F	Human	CGGTTTGTTTGGGTTTGGGTTTGGGTTTGGGTTTGGGT	75 bp
	Telo R	Human	GGCTTGCCTTACCCTTACCCTTACCCTTACCCTTACCCT	75 bp
	36B4 F	Human	CAGCAAGTGGGAAGGTGTAATCC	82 bp
	36B4 R	Human	CCCATTCTATCATCAACGGGTACA	82 bp

**Table 2 cells-11-03300-t002:** Comparison of hybrid *TERRA* levels and telomere length in patients with different PKD mutations; data were summarized as medians (quartiles 1–3).

Variable	PKD Mutation		*p* Value
	PKD1 (*n* = 43)	PKD2 (*n* = 22)	
***TERRA* Levels**	1.69(1.04–3.36)	1.15(0.97–2.21)	0.039
**Telomere Length**	0.14(0.11–0.28)	0.21(0.12–0.56)	0.126

**Table 3 cells-11-03300-t003:** Comparison of clinical variables between normal and rapidly progressive patients with ADPKD; data were summarized as *n* (%), mean ± standard deviation or median (1st–3rd quartiles). Significant *p* values are shown in bold.

Variable	Groups		Total	*p* Value
	Normal Progression (*n* = 62)	Rapid Progression (*n* = 16)		
**Age (year)**	49.26 ± 13.43	44.63 ± 14.29	48.31 ± 13.64	0.228
**Sex**	32(51.6)	9(56.3)	41(52.6)	0.741
**Follow-up time (months)**	84.0(45.0–96.0)	96.0(54.0–96.0)	84.0(48.0–96.0)	0.258
**PKD gene mutation**				
** *PKD1* **	31(50.0)	12(75.0)	43(55.1)	0.078
** *PKD2* **	21(33.9)	1(6.3)	22(28.2)	
**Non-diagnosed**	10(16.1)	3(18.8)	13(16.7)	
**eGFR (mL/min/1.73 m^2^)**	77.00(47.00–91.00)	35.50(25.50–54.25)	67.00(35.50–89.00)	<0.001
**Sodium (mmol/L)**	138.43 ± 3.69	136.81 ± 3.49	138.09 ± 3.69	0.119
**Potassium (mmol/L)**	4.38 ± 0.52	4.74 ± 0.74	4.46 ± 0.59	0.027
**Albumin (g/dL)**	4.31 ± 0.42	4.28 ± 0.37	4.30 ± 0.41	0.774
**Glucose (mmol/L)**	201.5(156.5–273.5)	258.0(189.0–290.0)	208.0(160.0–288.0)	0.317
**Triglyceride (mg/dL)**	96.0(89.0–102.0)	97.5(88.5–152.0)	96.0(89.0–104.5)	0.321
**LDL cholesterol (mg/dL)**	124.90 ± 29.75	126.00 ± 43.24	125.18 ± 33.19	0.931
**Hemoglobin(g/dL)**	13.45 ± 1.81	12.99 ± 1.48	13.35 ± 1.74	0.349
**CRP (mg/dL)**	3.00(3.00–5.00)	13.00(7.00–18.00)	4.00(3.00–7.00)	<0.001
**Proteinuria (g/dL)**	0.17(0.10–0.25)	0.86(0.56–1.20)	0.21(0.10–0.50)	<0.001
**Hybrid *TERRA* levels**	1.20(0.84–1.67)	2.70(1.61–4.00)	1.31(0.93–2.11)	<0.001
**Telomere length**	0.23(0.12–0.73)	0.12(0.07–0.41)	0.22(0.12–0.71)	0.041

**Table 4 cells-11-03300-t004:** Results of binary logistic regression analyses to predict rapid progression in ADPKD patients. OR: odds ratio, CI: confidence interval. Significant *p* values are shown in bold.

Variable	Univariate		Multiple	
	OR (%95CI)	*p* Value	OR (%95CI)	*p* Value
**Age (year)**	0.97(0.93–1.02)	0.227	-	-
**Sex (female/male)**	1.21(0.40–3.64)	0.001	-	-
**Follow-up time (months)**	1.01(0.99–1.03)	0.349	-	-
**PKD gene mutation**				
** *PKD1* **	1.00	-	-	-
** *PKD2* **	1.29(0.30–5.51)	0.731	-	-
**Non-diagnosed**	0.16(0.02–1.72)	0.130	-	-
**eGFR (mL/min/1.73 m^2^)**	0.97(0.94–0.99)	0.002	-	-
**Sodium (mmol/L)**	0.89(0.77–1.03)	0.122	-	-
**Potassium (mmol/L)**	3.07(1.10–8.58)	0.032	-	-
**Albumin (g/dL)**	0.82(0.21–3.15)	0.771	-	-
**Glucose (mmol/L)**	1.00(1.00–1.01)	0.498	-	-
**Triglyceride (mg/dL)**	1.03(1.01–1.05)	0.014	-	-
**LDL cholesterol (mg/dL)**	1.00(0.98–1.02)	0.914	-	-
**Hemoglobin(g/dL)**	0.86(0.63–1.18)	0.347	-	-
**CRP (mg/dL)**	1.08(1.01–1.17)	0.029	1.21(1.05–1.38)	0.007
**Proteinuria (g/dL)**	91.11(9.75–851.49)	<0.001	16.64(1.13–245.47)	0.041
***TERRA* levels**	3.67(1.74–7.74)	0.001	5.60(1.46–21.51)	0.012
**Telomere length**	0.48(0.10–2.29)	0.157	-	-

**Table 5 cells-11-03300-t005:** ROC analysis results for *TERRA* levels in predicting rapid progression in ADPKD patients; AUC: area under the curve, ROC: receiver operating characteristic, CI: confidence interval.

Potential Biomarker	ROC statistic		Diagnostic Statistic			
	AUC (%95 CI)	*p* Value	Sensitivity	Specificity	Positive Predictive Value	Negative Predictive Value
***TERRA* Levels <1.804**	0.836 (0.725–0.947)	<0.001	0.75 (0.48–0.93)	0.79 (0.67–0.88)	0.48 (0.33–0.80)	0.93 (0.79–0.96)

## Data Availability

Not applicable here.

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
