# Peer review of "Predicting Progression of Autosomal Dominant Polycystic Kidney Disease by Changes in the Telomeric Epigenome"

_cells, 2022, doi:10.3390/cells11203300_

Round 1
Reviewer 1 Report
The authors provided a new angle to study ADPKD. This is the first paper I have read that link ADPKD with telomere. The authors found that TERRA levels are negatively correlated with rapid form of ADPKD. I think the novelty is enough and the paper should be acknowledged after addressing the following issues.
Major
1. The abstract and introduction parts should be more informative and fluent. It is rather strange to notice the "nucleoprotein" sentence in the abstract without proper description for an outsider of telomere field. There is lack of illustration how the authors found it is necessary to study telomere in ADPKD patient in the Introduction. Additionally, there is no brief description of the result in the Introduction part.
2. Material and methods section should be detailed. E.g., it is inappropriate to say "under appropriate PCR conditions". What kind of condition? Another issue is that there is no statistics subsection.
3. The figures are poorly organized. There are only 'A' and 'B' in figure 1, what about the lower panel of figure 1?
4. The figure legends are too concise, which could not depict the contents in the figures. And how the data are presented? mean and SD, mean and SEM, or median and IQR?
5. From a personal view, the linear models used in the study should be shown in mathematical forms, other than just verbal description. E.g., multivariate model, what kind of model?
6. The authors used qRT-PCR to detect TERRA activity in DNA-RNA hybrid form. Is there any method to detect the total amount of TERRA or just the 'free' TERRA? Is it necessary to do this?
7. The author presented that TERRA is a 'biomarker' for rapid ADPKD, but without testing its level in a testing group. It is more complete to test whether TERRA activity is elevated in a new ADPKD group.
Minor
1. Line 179. 'Multivariate'. I think it is a common ambiguity for all scientists even the statisticians. In the paper, the authors used a 'multiple regression model' rather than 'multivariate model'.
2. Line 107. 'sequence of which is' or 'sequences of which are'?
3. Line 116, 118, 123, and 125. 'poured' or 'discarded'?
4. Line 160. 'DNase' and 'enzymatic'.
5. Line 146. How many patients were enrolled in the study?
6. Table 5. 'negative'.
7. the decimal points: use ',' or '.' should be consistent. Please refer to the requirement of mdpi publication.
Author Response
The authors provided a new angle to study ADPKD. This is the first paper I have read that link ADPKD with telomere. The authors found that TERRA levels are negatively correlated with rapid form of ADPKD. I think the novelty is enough and the paper should be acknowledged after addressing the following issues.
Major
- The abstract and introduction parts should be more informative and fluent. It is rather strange to notice the "nucleoprotein" sentence in the abstract without proper description for an outsider of telomere field. There is lack of illustration how the authors found it is necessary to study telomere in ADPKD patient in the Introduction. Additionally, there is no brief description of the result in the Introduction part.
Authors:
Thank you for comments, we have introduced changes. In the text all changes are in red.
- Material and methods section should be detailed. E.g., it is inappropriate to say "under appropriate PCR conditions". What kind of condition? Another issue is that there is no statistics subsection.
Authors:
We have changed to:
-Then, the desired product was placed in the Thermal Cycler device under reaction conditions for PCR. The thermal cycling conditions consisted of a step at 42°C for 15 min, followed 85°C for 5 min, 65°C for 15 min, and finally, cooled at 4°C for 5 min. At the end of PCR, 20µl of cDNA product was formed and cDNA samples were diluted 1:5 with nuclease-free water for PCR amplification.
-The Ct values of the samples were determined in a Roche Light Cycler LC 480 II Real-Time PCR. Real-Time PCR conditions consisted of an initial denaturation step at 95°C for 5 min, followed by 40 cycles of 94°C for 20 sec, 60°C for 20 sec, 72°C for 45 sec, and 95°C for 15 sec, and, finally, melting curve was performed at 67°C for 01 sec (melting curve) and cooled at 40°C for 30 sec. All samples were run in duplicate to rule out manipulation errors, and Ct values were averaged. β-Actin was used as the House-Keeping gene. CT values obtained at the end of PCR were normalized using the 2-ΔΔCt method [17, 18].
And
Statistical Analysis section is transferred from results and added under Materials section.
- The figures are poorly organized. There are only 'A' and 'B' in figure 1, what about the lower panel of figure 1?
Authors:
Thank you for comment, we have corrected.
- The figure legends are too concise, which could not depict the contents in the figures. And how the data are presented? mean and SD, mean and SEM, or median and IQR?
Authors:
Apologize, we added more information
- From a personal view, the linear models used in the study should be shown in mathematical forms, other than just verbal description. E.g., multivariate model, what kind of model?
Authors:
The used linear model is multiple binary logistic regression model which is explained in detail in Statistical Analysis section. The ‘multivariate’ word is replaced with ‘multiple’. The mathematical form of the obtained final model is given in Results section.
- The authors used qRT-PCR to detect TERRA activity in DNA-RNA hybrid form. Is there any method to detect the total amount of TERRA or just the 'free' TERRA? Is it necessary to do this?
Authors:
Trizol-chloform extracion releases cytoplasmic and nuclear RNA (almost total). However, there is also a fraction of RNA strongly attached to DNA (genomic) that are not released by Trizol-chloroform buffer, and are recovered only after Dnase treatment from DNA fraction.
- The author presented that TERRA is a 'biomarker' for rapid ADPKD, but without testing its level in a testing group. It is more complete to test whether TERRA activity is elevated in a new ADPKD group.
Authors:
We do not have a new separate group to test the TERRA activity. We added this limitation to the Conclusion section.
Minor
- Line 179. 'Multivariate'. I think it is a common ambiguity for all scientists even the statisticians. In the paper, the authors used a 'multiple regression model' rather than 'multivariate model'.
Authors:
The ‘multivariate’ word is replaced with ‘multiple’ as suggested.
- Line 107. 'sequence of which is' or 'sequences of which are'?
Authors:
We changed to:
Primer sequences are shown in Table 1.
- Line 116, 118, 123, and 125. 'poured' or 'discarded'?
Authors:
Sorry: discarded
- Line 160. 'DNase' and 'enzymatic'.
Authors:
Corrected
- Line 146. How many patients were enrolled in the study?
Authors:
78
- Table 5. 'negative'.
Authors:
Thanks, it is now corrected.
- the decimal points: use ',' or '.' should be consistent. Please refer to the requirement of mdpi publication.
Authors:
Thank you for the comment, we have corrected all to (.). Period (.) is used as decimal separators throughout the manuscript.
Reviewer 2 Report
Intra-familiar heterogeneity of phenotypes in ADPKD patients is a long-standing, unresolved issue. From anticipation to DNA methylation several factors has been claimed to explain this occurrence. Therefore, studies that addresses this issue are welcome and may be important for treatment planning with the new vasopressin receptors antagonists.
Authors proposed a new mechanism associated with changes in telomeric epigenome - shortened telomeres and increased expression levels of non-coding RNA (TERRA) in patients with ADPKD.
These are my notes on the manuscript:
Main concerns
1. We know that there are substantial differences in the presentation and progression of patients with PKD1 mutation as compared to PKD2 mutation. Therefore, authors should not include in the same analysis PKD1 and PKD2 patients, even assuming that the similar findings could explain variability in clinical findings in families with PKD1 and PKD2 mutations.
2. Intra-familiar variability is one issue in clinical management of ADPKD patients. However authors did not present their results to help in resolving this issues. How was the variability of telomeres and TERRA within members of the same family?
2. Authors refer that shortened telomeres is a distinct feature of CKD which means that a control group with CKD could be included to ascertain whether altered telomere length is unique to ADPKD.
3. I suggest English editing services to review all the manuscript.
Minor aspects
1. Beyond PKD1 and PKD2 genes, mutations in GANAB gene is now recognized as responsible for a minority of patients with ADPKD.
2. Number of included patients varies between 79 (abstract; line 72) and 78 (results – line 146)
3. Clinical characteristics of the patients is poor– how many are PKD1, PKD2 or without genetic characterization (a glimpse of this information appears in table 1 and 2)? How is their renal function at the time they were studied? Did transplanted patient be included? How many were classified as rapid progressors and slow progressors?
4. Sometimes there is confusion between presentation of results and discussion
a. Line 165/166 – “It is speculated that there is … “ – this should come in the discussion section, not under the results section.
5. Some sentences are poorly formulated:
a. “Rapidly progressing patients were compared with each other” – line 173 – what does this mean?
b. “Renal failure due to renal parenchyma resulting from cysts developing in the kidneys” – line 276
Author Response
Main concerns
- We know that there are substantial differences in the presentation and progression of patients with PKD1 mutation as compared to PKD2 mutation. Therefore, authors should not include in the same analysis PKD1 and PKD2 patients, even assuming that the similar findings could explain variability in clinical findings in families with PKD1 and PKD2 mutations.
Authors:
We separately presented PKD1 and PKD2 patients as well as males and females samples analysis in Figure 2.
- Intra-familiar variability is one issue in clinical management of ADPKD patients. However authors did not present their results to help in resolving this issues. How was the variability of telomeres and TERRA within members of the same family?
Authors:
In the new Figure 4, we have detailed the family variations of our cohort which includes 24 members of 11 distinct families.
- Authors refer that shortened telomeres is a distinct feature of CKD which means that a control group with CKD could be included to ascertain whether altered telomere length is unique to ADPKD.
Authors:
We currently do not have new groups of CKD patienst to test for telomere length and TERRA activity.
After this study, we plan to determine the profiles of patients who have CKD for different reasons. We will thus determine whether the data obtained are specific to ADPKD or CKD.
- I suggest English editing services to review all the manuscript.
Authors:
This manuscript was English edited by our colleagues Kenneth Marcu (Stony Brook University), none of the author are native of English language, if our manuscript is accepted we will again edit by English editing service of the journal.
Minor aspects
- Beyond PKD1 and PKD2 genes, mutations in GANAB gene is now recognized as responsible for a minority of patients with ADPKD.
Authors:
We have not yet studied GANAB gene, very good suggestion we could plan for further studies with new group and after agreement of the patients.
- Number of included patients varies between 79 (abstract; line 72) and 78 (results – line 146)
Authors:
Sorry, the number of patients is 78. We corrected this throughout the manuscript.
- Clinical characteristics of the patients is poor– how many are PKD1, PKD2 or without genetic characterization (a glimpse of this information appears in table 1 and 2)? How is their renal function at the time they were studied? Did transplanted patient be included? How many were classified as rapid progressors and slow progressors?
Authors:
In our data, 62 of the patients were classified as normal progressors, while 16 of the patients were classified as rapid progressors.
- Sometimes there is confusion between presentation of results and discussion
- Line 165/166 – “It is speculated that there is … “ – this should come in the discussion section, not under the results section.
Authors:
Right, we have deleted from results section.
- Some sentences are poorly formulated:
- “Rapidly progressing patients were compared with each other” – line 173 – what does this mean?
- “Renal failure due to renal parenchyma resulting from cysts developing in the kidneys” – line 276
Authors:
We have changed to:
- Patients with fast evolution stand out from the rest; by shorter telomere length (p=0.041) and with higher levels of TERRA in hybrid structure with DNA (p<0.001). The comparative analysis and results are shown in Figure 2. Additionally, the characteristics of the two groups are summarized in Table 3.
b.
Renal failure due to parenchyma cysts development in the kidneys, as well as the onset of hypertension at an early age is a prominent feature of ADPKD.
Round 2
Reviewer 1 Report
I think the authors have addressed most of my concerns except some minor ones.
1. Since there is no validation group involved, the TERRA could not be a 'biomarker'. I suggest using 'potential biomarker' instead.
2. There are still some typos or nomenclature/italic issues regarding proteins and genes throughout the manuscript.
Author Response
Reviewer 1
Comments and Suggestions for Authors
I think the authors have addressed most of my concerns except some minor ones.
-Authors
We thank for your time and attention.
Reviewer 1
- Since there is no validation group involved, the TERRA could not be a 'biomarker'. I suggest using 'potential biomarker' instead.
-Authors
We agree word changes are included.
Reviewer 1
- There are still some typos or nomenclature/italic issues regarding proteins and genes throughout the manuscript.
-Authors
Sorry, for the errors, we have corrected I hope everything is corrected.
Reviewer 2 Report
I would like to thank authors for taking into account my suggestions. The only question that still remain, in my opinion, is related to clear separation between PKD1 and PKD2 patients as, it seems to me, that cannot be analyzed together.
Looking at the results in table 2, we acknowledge that PKD2 patients, as expected, were predominantly ‘normal progressors’. Therefore, analysis of TERRA levels should take in account this subset of patients: comparisons should only include ‘normal progressors’ versus ‘rapid progressors’ in both PKD genes.
Author Response
Reviewer 2
I would like to thank authors for taking into account my suggestions. The only question that still remain, in my opinion, is related to clear separation between PKD1 and PKD2 patients as, it seems to me, that cannot be analyzed together.
-Authors
We thank you for your time and attention and we apologize if we have not fully answered your question, it may have been because it was not clear to us. It is also our fault if we did not specify that all the patients of the "rapid progression" are with PKD1 except one with the PKD2 mutation.
Reviewer 2
Looking at the results in table 2, we acknowledge that PKD2 patients, as expected, were predominantly ‘normal progression’. Therefore, analysis of TERRA levels should take in account this subset of patients: comparisons should only include ‘normal progression’ versus ‘rapid progression’ in both PKD genes.
-Authors
In our group of patients, all patients with rapidly progressing ADPKD have a PKD1 mutation, all but one with rapidly progressing have a PKD2 mutation, and the others have normal progression. For this reason, a graph showed the telomere and TERRA levels of patients with the PKD2 mutation with slow and rapid progression but the results could not be statistically assessed.